# Distilling ChatGPT for Explainable Automated Student Answer Assessment

**Jiazheng Li[1], Lin Gui[1], Yuxiang Zhou[1], David West[2], Cesare Aloisi[2] and Yulan He[1,3]**

[1]Department of Informatics, King's College London, UK
[2]AQA, UK    [3]The Alan Turing Institute, UK
{jiazheng.li, lin.1.gui, yuxiang.zhou, yulan.he}@kcl.ac.uk
{caloisi, dwest}@aqa.org.uk

## Abstract

Providing explainable and faithful feedback is crucial for automated student answer assessment. In this paper, we introduce a novel framework that explores using ChatGPT, a cutting-edge large language model, for the concurrent tasks of student answer scoring and rationale generation. We identify the appropriate instructions by prompting ChatGPT with different templates to collect the rationales, where inconsistent rationales are refined to align with marking standards. The refined ChatGPT outputs enable us to fine-tune a smaller language model that simultaneously assesses student answers and provides rationales. Extensive experiments on the benchmark dataset show that the proposed method improves the overall QWK score by 11% compared to ChatGPT. Furthermore, our thorough analysis and human evaluation demonstrate that the rationales generated by our proposed method are comparable to those of ChatGPT. Our approach provides a viable solution to achieve explainable automated assessment in education[1].

## 1 Introduction

Student answer assessment is a critical component of the education process. Prompt and insightful answer assessments can enhance students' learning experiences and support their academic growth (Nicol and Macfarlane-Dick, 2006). However, manually providing detailed feedback is time-consuming, and differences in assessment criteria among various evaluators can result in inconsistencies in the grading process (Weigle, 2002).

Various automated student answer assessment models have been proposed in recent years, mostly built on the Pre-trained Language Models (PLMs) (Devlin et al., 2019), making the assessment process more efficient and consistent. These approaches (Sung et al., 2019; Mayfield and Black,

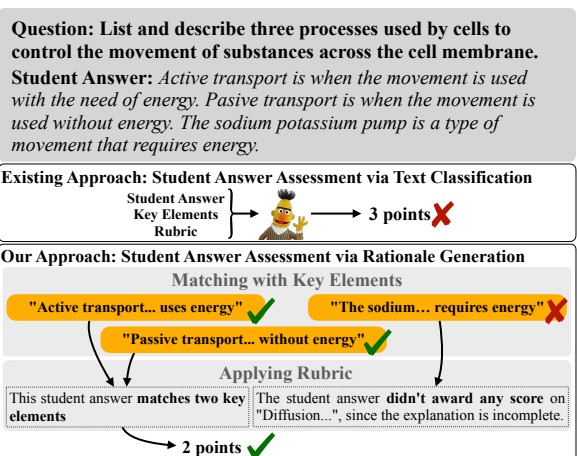

Figure 1: Classification-based automated answer assessment techniques often rely on black-box models, making the assessment process challenging to interpret. Incorporating rationale generation can significantly enhance the transparency of the assessment decisions.

2020) tend to frame the assessment task as a classification problem, which involves training text classifiers to predict scores given student answers. However, as shown in Figure 1, the feedback provided in terms of scores is not sufficiently detailed for students to identify weaknesses in their answers. Besides, it is challenging for humans to interpret the classifiers' decision-making process, making classifiers' assessment results less trustworthy.

Researchers have advocated for generating rationales to enhance the interpretability of classifiers. These rationales are natural language explanations that substantiate model predictions (Gurrapu et al., 2023). Often, such strategies necessitate rationale annotations on classification datasets for effective training (Camburu et al., 2018). However, most available datasets in student answer assessments only include score annotations. Providing detailed rationale annotation on existing datasets requires significant domain expert efforts. Furthermore, rational annotations are constrained by the specificity of the information in the dataset, making it difficult

---

[1]Code available at https://github.com/lijiazheng99/aera.

to generalise across diverse academic subjects.

Recent developments on Large Language Models (LLMs), including ChatGPT (Stiennon et al., 2020), have demonstrated impressive capabilities in various Natural Language Processing (NLP) applications. For example, these models have exhibited remarkable performance in arithmetic and common sense reasoning while showing their potential for performing step-by-step reasoning (Wei et al., 2022). Furthermore, Gilardi et al. (2023) found that using ChatGPT for data annotation outperforms crowd workers with much lower costs. It becomes possible to improve the interpretability of student answer assessment, by harnessing the capabilities of LLMs without relying on expensive human annotation processes. However, LLMs' running costs, non-open-source issues and limited specialization still hinder their applications.

This paper introduces the **AERA** (**A**utomated **E**xplainable Student **R**esponse **A**ssessment) framework, designed to harness ChatGPT as a reasoning teacher. The aim is to distil a more compact language model, enabling it to produce rationales and enhance interpretability on student answer assessment. We first designed several prompt templates with different levels of instruction to examine ChatGPT's capabilities on student answer assessment and rationale generation. Then, we enhance the quality of rationales with a rationale refinement module. Last, a smaller language model is fine-tuned on the refined data to perform the answer assessment and rationale generation. Since there are no established automatic metrics to evaluate the correctness of rationales without ground truth annotations, we conducted a comprehensive human evaluation, assessing the rationales generated by **AERA** and comparing them with those generated by ChatGPT. Our experimental results show that, within our designed framework, a smaller language model can surpass ChatGPT in terms of assessment performance while generating more accurate rationales to explain the assessment decision.

In summary, our contributions are: (1) We proposed a framework **AERA**, to distil the rationale generation capability of ChatGPT into a smaller language model; (2) We introduced two strategies for ChatGPT to refine its rationales independently; (3) Through comprehensive experiments and human evaluation, we show that our method is able to generate high-quality rationales without the need of additional annotation for model learning. To the best of our knowledge, AERA is the pioneering framework that leverages ChatGPT to generate rationales for explainable student answer assessments using more compact language models.

## 2 Related Work

**Automated Student Answer Assessment** Also known as automated essay scoring, where most researchers model the problem as a text classification task (Uto, 2021). Early approaches (Alikaniotis et al., 2016; Dong et al., 2017) built on deep neural networks shed new light on efficient and consistent assessment solutions. Recent advents in PLMs (Devlin et al., 2019; Brown et al., 2020) provide better text representations to develop more accurate PLM-based scoring systems (Mayfield and Black, 2020; Yang et al., 2020). Nevertheless, limited knowledge of the assessment system's decision-making process raised concerns about its fairness and usefulness. Alikaniotis et al. (2016); Yang et al. (2020) tried to improve assessment interpretability via attention mechanisms. Filighera et al. (2022) annotated a student feedback dataset for more explainable assessment results with feedback.

**Rationale Generation in Text Classification** Generate rationales for text classifiers have gained increasing attention due to concerns in interpretability (Gurrapu et al., 2023; Li et al., 2023a). Researchers tried to generate rationales on various tasks, including sentiment analysis (Antognini and Faltings, 2021), review classification (Liu et al., 2019), and natural language inference (Camburu et al., 2018). Those approaches mainly fall into two categories: extractive rationale generation (Lei et al., 2016), where rationales are extracted from the input features; and abstractive rationale generation (Marasovic et al., 2022), where rationales are paraphrased from existing sentences or newly generated. LLMs showcased the great potential to use their in-context learning ability for abstractive rationale generation (Marasovic et al., 2022), which provides a viable solution for our task.

In our study, we tackle the interpretability challenge in automated student answer assessments by producing abstractive rationales from ChatGPT and distilling a smaller model to perform the same task.

## 3 AERA: Automated Explainable Student Response Assessment Framework

Applications of student answer assessment systems built on PLMs have been hindered by concerns

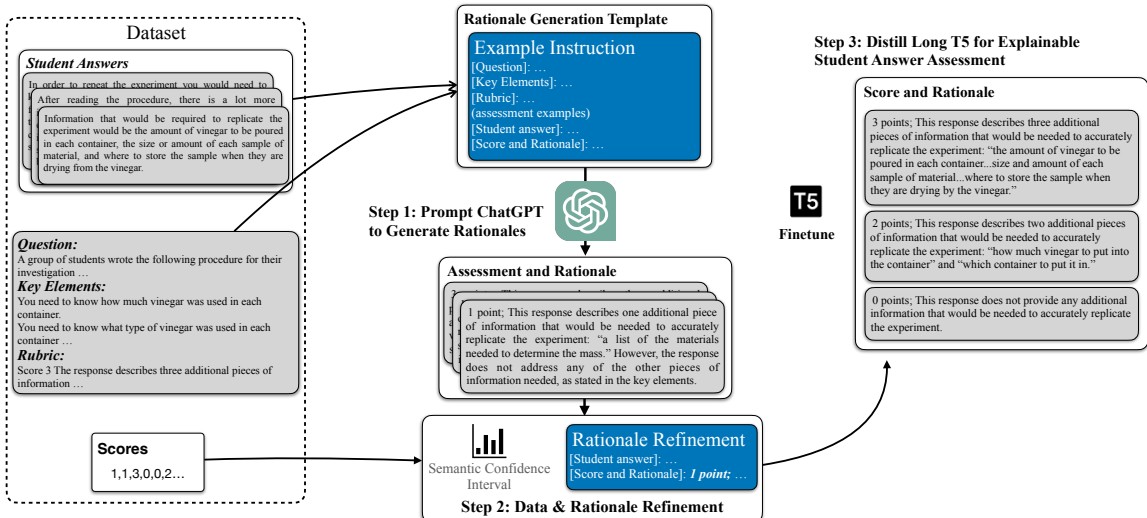

Figure 2: **AERA** framework contains three steps: (1) Prompting ChatGPT for rationale generation; (2) Applying rationale refinement strategies to improve the quality of the rationales; and (3) Distilling a smaller language model for more efficient rationale generation.

about their interpretability. Existing explanation methods built on classification-based assessment systems struggle to provide natural language explanations for their decision-making process, thus making their application less useful for education purposes. Additionally, the limited availability of datasets annotated with grading rationales, coupled with the substantial expenses of human annotation, poses significant obstacles to the advancement of rationale generation approaches.

To address the above challenges, we introduce **AERA** framework, which leverages the in-context learning capabilities of LLMs to generate rationales and fine-tune smaller language models for explainable student answer scoring. As shown in Figure 2, our approach consists of three main steps: (1) We design various prompt templates according to different levels of reasoning difficulties and instruct ChatGPT to assess student answers while providing rationales. (2) It is important to acknowledge that ChatGPT may not be able to assess all student answers accurately. To address this limitation, we introduce a rationale and data refinement module which aims to enhance the quality and usability of the generated rationales. (3) The generated rationales, despite the presence of noise, can be utilized to efficiently fine-tune smaller language models, enabling the generation of plausible rationales for student answer assessment.

**Problem Setup**   A typical student answer assessment dataset includes five components. The ques-

tion $\mathcal{Q}$[2]; Key elements $\mathcal{K}$ that list the expected key answer elements; Rubrics $\mathcal{R}$, a grading guide used to evaluate the quality of student answers[3]; A collection of student answers $X = \{x_i\}_{i=1}^{N}$; and a collection of the corresponding scores $Y = \{y_i\}_{i=1}^{N}$. When preparing the key elements and rubric used for assessment, lead examiners will also provide sample answer assessments during standardisation meetings[4]. We denote those sampled student answers, scores and grading rationale as $(x'_j, y'_j, r'_j), j = 1, 2, \cdots, M$. For a given student answer $x_i$, we use $\hat{y}_i$ to denote the predicted score and $\hat{r}_i$ to denote the generated rationale.

We use the following notations to describe the model generation process: $X \rightarrow Y$, where the model directly predicts a score given a student answer; $X \rightarrow YR$, where the model predicts a score and generates a rationale given a student answer; $XY \rightarrow R$, where both a student answer and its corresponding score are given to the model to generate a rationale. For the rest of the section, we highlighted examples from sample assessment in green and models' output in blue.

### 3.1 Prompting ChatGPT for Rationale Generation

Recent advances in ChatGPT showcased its great potential to generate rationales on complex reason-

---

[2]Some questions contain tabular data. We leverage ChatGPT's table-understanding capability to create table descriptions from tabular data in A.1.

[3]See §C for detailed questions, key elements and rubrics.

[4]A standardisation guide from a leading exam service.

ing tasks, such as arithmetic computation. However, student answer assessment is a complex decision-making process in education involving various reasoning phases (Bejar, 2012). The main challenges for an assessment task include finding the valid key elements stated in the student's answer and deciding a proper score range that applies to the answer. To the best of our knowledge, there is scarce work researching the viable prompting strategy for student answer assessment with ChatGPT. Following the taxonomy from Karmaker and Feng (2023), we propose three prompt templates for rationale generation at different reasoning levels to explore ChatGPT's reasoning capability and identify the prompt leading to more accurate assessment results. We begin with the Simple Instruction template, and progressively reduce the level of reasoning difficulty by incorporating more elaborate natural language instructions or patterns extracted from assessment samples into the template.

**Simple Instruction**  We first use a simple $X \to YR$ prompt instruction that only contains a single question to ask ChatGPT to elaborate the reason for its scoring process:

```
[Question]: <𝒬>
[Key Elements]: <𝒦>
[Rubric]: <ℛ>
[Student answer]: <x_i>
What score should this Student answer
get and why?
(free form of r̂_i and ŷ_i)
```

Listing 1: Simple Instruction Prompt Template

Given the intricate nature of the student answer assessment task, this prompt presents the highest level of difficulty. ChatGPT needs to plan its assessment cycle, understand the meaning of key elements and the rubric, and appropriately execute the assessment to match the student answer with the key elements and apply the rubric for scoring and rationale generation.

**Complex Instruction**  Previous research suggests that more elaborate natural language prompt instruction may improve the reasoning capabilities of LLMs (Karmaker and Feng, 2023; Brown et al., 2020). Therefore, we design a more detailed $X \to YR$ prompt instruction that clearly outlines the functionality of key elements and the rubric and provides clear guidance on how to apply them in student answer assessment:

```
[Question]: <𝒬>
[Key Elements]: <𝒦>
```

```
[Rubric]: <ℛ>
[Student answer]: <x_i>
Carefully read the [Question], [Key
Elements], and [Rubric], then compare [
Student answer] with the [Key Elements],
 and apply the [Rubric] to derive the
student score. Please be certain to
spell out your reasoning so anyone can
verify them. Spell out the [Key Elements
] that the [Student answer] matches, and
 also spell out which rule in the [
Rubric] is applied.
(free form of r̂_i and ŷ_i)
```

Listing 2: Complex Instruction Prompt Template

Compared with the *Simple Instruction* template, the *Complex Instruction* template offers additional guidance in the assessment process, thereby reducing the level of difficulty.

**Example Instruction**  Although ChatGPT has demonstrated impressive reasoning capabilities in understanding natural language instructions, it faces some limitations when employing zero-shot based templates such as the aforementioned *Simple* and *Complex Instructions*. Specifically, it tends to generate free-form rationales that require additional annotations for score and rationale extraction. In addition, it also suffers from hallucination problems. Previous research (Brown et al., 2020; Marasovic et al., 2022; OpenAI, 2023; Karmaker and Feng, 2023) has shown the benefits of few-shot based templates, as they allow output formatting through examples, eliminating the need for annotations for score extraction. Furthermore, leveraging the patterns from demonstration examples, LLMs can achieve better performance. To this end, we proposed a $X \to YR$ example instruction prompt, utilizing the sample answer assessments obtained from standardisation as demonstration examples, for generating properly formatted rationales:

```
[Question]: <𝒬>
[Key Elements]: <𝒦>
[Rubric]: <ℛ>
[Student answer]: <x'_1>
[score and Rationale]: <y'_1>; <r'_1>
...(assessment examples)
[Student answer]: <x'_M>
[score and Rationale]: <y'_M>; <r'_M>
[Student answer]: <x_i>
[score and Rationale]: <ŷ_i>; <r̂_i>
```

Listing 3: Example Instruction Prompt Template

## 3.2  Data & Rationale Refinement

Given the lack of established approach to evaluate the correctness of generated rationales without

gold annotation, we follow a previous study (Ho et al., 2023) by assuming the rationale supports the score if the LLM-predicted answer score is correct. However, it is important to note that ChatGPT cannot guarantee the correctness of all the assessed scores on the whole dataset. Incorrect predictions can arise from two scenarios: (1) The dataset contains wrongly labelled score; or (2) ChatGPT's predictions are wrong. To address these situations, we introduce refinement strategies to improve the rationale generation's success rate.

**Fixing Wrongly Labelled Data**   ChatGPT, being a non-deterministic language model, can generate varying outputs with each iteration. We utilise the semantic confidence interval for LLMs outlined by Kuhn et al. (2023) to calculate the uncertainty of scores associated with the generated rationales. Based on our observation, generated rationales $\hat{r}_i$ that correspond to the same assessed score $\hat{y}_i$ are semantically similar. Therefore, the predictive probability of each assessed score $\hat{y}_i$ can be represented as: $p(\hat{y}_i \mid x_i) = \sum_{\hat{y}_i \in S} p(\hat{y}_i \mid x_i)$; where $S$ is the set of all occurrences of semantically similar rationales shares the same predicted score.

Through our experiments, we demonstrate that gold annotations might be wrong for highly confident incorrect assessments made by ChatGPT, when the score difference exceeds one. This approach helps to identify corrupted input data and human labelling errors, ultimately reducing data uncertainty and improving overall data quality.

**Prompt for Rationale Refinement**   Since the $X \rightarrow Y R$ prompt cannot guarantee the correctness of the score, we introduce a $XY \rightarrow R$ rationale refinement template. This template is based on the *Example Instruction* prompt template and incorporates a given score as input, LLM can use the score as prior knowledge to locate a proper distribution that generates more accurate rationales:

```
[Question]: <Q>
[Key Elements]: <K>
[Rubric]: <R>
[Student answer]: <x'_1>
[score and Rationale]: <y'_1>; <r'_1>
...(assessment examples)
[Student answer]: <x'_E>
[score and Rationale]: <y'_E>; <r'_E>
[Student answer]: <x_i>
[Score and Rationale]: <y_i>; <r̂_i>
```

Listing 4: Rationale Refinement Prompt Template

## 3.3   Distilling Student Model for Efficient Rationale Generation

Although LLMs have exhibited impressive in-context learning and reasoning capabilities, huge parameter size, non-open source issues, and enormous running costs (Independent, 2023; Li et al., 2023b) make them hard to be developed and trained locally. Besides, uncontrollable, occasionally unexpected outputs (e.g. hallucination) render LLMs less practical for real-world student answer assessment. Consequently, we propose using ChatGPT-generated rationales to fine-tune a smaller language model for efficient rationale generation. Unlike previous literature that has focused on knowledge distillation in arithmetic chain-of-thought (Ho et al., 2023; Fu et al., 2023; Magister et al., 2023), student answer assessment is a much more complex reasoning task based on the input source (e.g. the scope of key elements and the definition of the rubric).

We utilise the rationales generated by ChatGPT, as described in §3.1, with their quality improved by fixing wrongly labelled data and further refinement outlined in §3.2, as training data for task-specific knowledge distillation. We adopt Long T5 (Guo et al., 2022) as our base model as T5 is one of the popular open-source PLM that has been pre-trained with many supervised tasks, including both classification and generation. Besides, prompt for student answer assessment is relatively long, Long T5 is capable of taking longer input than commonly used base models while maintaining little performance drop. Our fine-tuning process takes Question, Key Elements, Rubric, and student answer as input to predict the score and generate rationale, $X \rightarrow Y R$. Prompt template used for fine-tuning is as follows:

```
[Question]: <Q>
[Key Elements]: <K>
[Rubric]: <R>
[Student answer]: <x_i>
[Score and Rationale]: <ŷ_i>; <r̂_i>
```

Listing 5: Prompt Template for Fine Tuning

## 4   Experiments

### 4.1   Experimental Setup

**Dataset**   We employ the Hewlett Foundation: Short Answer Scoring (ASAP-SAS) dataset[5]. This dataset encompasses over 23,000 short answer responses from students in grades 7 to 10, including

[5]https://kaggle.com/competitions/asap-sas

| Dataset (Subject) | #1 (Science) | | | #2 (Science) | | | #5 (Biology) | | | #6 (Biology) | | | Overall | | |
|---|---|---|---|---|---|---|---|---|---|---|---|---|---|---|---|
| Method/Model | Acc | F1 | QWK | Acc | F1 | QWK | Acc | F1 | QWK | Acc | F1 | QWK | Acc | F1 | QWK |
| $X \rightarrow Y$ Fine-tuned Text Classification | | | | | | | | | | | | | | | |
| BERT | 66.79 | 67.54 | 79.17 | 54.23 | 51.53 | 68.53 | 84.28 | 45.82 | 72.87 | 88.43 | 54.76 | 80.30 | 73.43 | 54.91 | 75.22 |
| Longformer | 74.15 | 74.81 | **83.75** | 62.75 | 63.21 | **78.79** | 83.67 | 58.02 | 80.63 | 88.09 | 59.44 | **83.25** | 77.17 | 63.87 | **81.61** |
| Longformer-all | 72.59 | 73.61 | 83.05 | 59.08 | 59.52 | 76.76 | 86.23 | 61.50 | **82.17** | 87.59 | 55.82 | 82.56 | 76.37 | 62.61 | 81.14 |
| $X \rightarrow YR$ ChatGPT Prompting | | | | | | | | | | | | | | | |
| Simple Instruction | 49.19 | 46.19 | 58.69 | 46.86 | 43.49 | 56.11 | 53.01 | 41.48 | 42.76 | 43.91 | 29.61 | 41.14 | 48.24 | 40.19 | 49.68 |
| Complex Instruction | 55.30 | 55.28 | 65.38 | 38.82 | 38.33 | 45.06 | 71.24 | 41.26 | 52.94 | 70.78 | 52.06 | 64.73 | 59.04 | 46.73 | 57.03 |
| Example Instruction | 55.66 | 53.75 | 61.40 | 49.06 | 48.12 | 63.20 | 68.06 | 54.02 | 68.17 | 68.45 | 50.39 | 64.66 | 60.31 | 51.57 | 64.36 |
| $X \rightarrow YR$ Fine-tuned Long T5 Rationalization | | | | | | | | | | | | | | | |
| **AERA(Ours)** | 63.26 | 62.90 | **75.06** | 43.27 | 42.35 | **54.15** | 83.78 | 53.29 | **76.44** | 89.37 | 60.38 | **80.81** | 69.92 | 54.73 | **71.62** |
| w/o Fixing Wrong Labels | 52.42 | 50.24 | 60.66 | 40.45 | 35.29 | 44.26 | 66.78 | 50.76 | 63.65 | 68.00 | 40.94 | 62.54 | 55.91 | 44.31 | 57.78 |
| w/o Rationale Refinement | 52.06 | 49.46 | 58.95 | 40.85 | 39.09 | 49.80 | 61.65 | 46.47 | 60.18 | 66.94 | 41.18 | 62.05 | 55.38 | 44.05 | 57.75 |
| Correct Score Only | 56.79 | 55.95 | 69.96 | 35.60 | 24.02 | 23.94 | 78.76 | 48.79 | 71.83 | 84.19 | 54.93 | 79.17 | 63.84 | 45.92 | 61.23 |

Table 1: Comparison of performance across classification baselines and rationale generation approaches. The highest QWK has been highlighted in **Bold** for fine-tuned models and underlined for LLM inference results.

ten questions spanning subjects such as Science, Biology, English, and Art. We only use four subsets focusing on Science and Biology questions.

**Baselines**  We compare our method with three classification setups: **BERT**: Bert-base-uncased model fine-tuned with student answers as inputs and scores as output (Mayfield and Black, 2020); **Longformer**: Longformer-base-4096 fine-tuned with student answers as input and scores as output; and **Longformer-all**: Longformer-base-4096 fine-tuned with the concatenation of additional information (question, key elements, rubric) and student answers as input and scores as output.

**Evaluation Metric**  We adopt the Accuracy (Acc) and macro f1 score (F1) and Quadratic Weighted Kappa (QWK) to evaluate the classification performance. We use sacreBLEU (Post, 2018) to measure the rationales' semantic similarity on the validation set and select the best checkpoint.
We provide detailed dataset description, QWK implementation and hyper-parameters setup in §B.1.

## 4.2 Overall Comparison

Table 1 displays the performance of student answer assessment across three task scenarios: fine-tuned text classification, ChatGPT prompting, and fine-tuned Long T5 for rationale generation.

For text classification baselines, when comparing BERT and Longformer, we observe that using a model that accommodates longer input text length can improve performance when trained solely on student answers. However, we do not see an improvement in overall performance when incorporating additional information, such as question,

key answer elements and rubric, into the input, which suggests that the text classifier may make predictions based on spurious features rather than checking student answer against the key answer elements and applying the supplied rubric. Hence, even though text classifiers may exhibit relatively high-performance scores, there remains a concern about the trustworthiness of their outputs.

For assessment and rationale generated from ChatGPT, we observe that the prompting under the few-shot setting (*Example Instruction*) is superior to the zero-shot settings (*Simple & Complex Instruction*), which achieved the highest overall performance with lower variances across four datasets.

Once we identified the viable prompt strategy for rationale generation, we fine-tuned Long T5 on the generated rationale for explainable student answer assessment. Our **AERA** framework obtained the highest overall performance compared with other rationale generation methods. Although the overall performance does not match that of text classifiers, given the intricate nature of the text generation task, noteworthy performance gains are observed on datasets #5 and #6, surpassing those achieved by the BERT classifier. This shows the benefit of enhancing the transparency of automated student answers assessment by generating rationales.

We conducted ablation studies to examine the effectiveness of each component in the Data & Rationale Refinement module in our framework. We find that if we only keep a subset of rationales with correctly predicted scores[6] (*Correct Score Only*), the performance on datasets #1, #5 and #6 surpasses those achieved when incorporating any of the addi-

---

[6]See §B.4 for detailed data statistic and other comparisons.

tional refinement strategies. Although these results show the strong performance brought by rationales with correctly predicted scores, this method may not be universally applicable when the amount of data is limited, as seen in dataset #2. After incorporating the two strategies separately, namely Fixing Wrong Labels & Rationale Refinement, to compose an updated dataset, we observed a significant performance improvement on dataset #2 due to the availability of more data. However, we see a performance drop on #1, #5 and #6 when compared with *Correct Score Only*, indicating the presence of wrongly labelled data or incorrectly predicted rationales can adversely impact the overall performance. In sum, both the data & rationale refinement components are essential within our framework to prevent data scarcity and effectively reduce noisy data.

### 4.3 Human Evaluation

We carried out two distinct human evaluations for rationales generated by both AERA and ChatGPT[7].

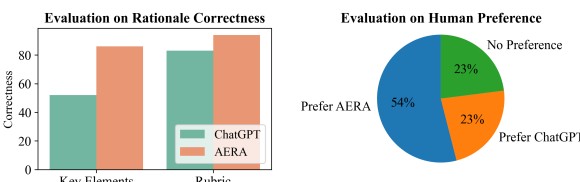

Figure 3: Visualization of Human Evaluation Results

**Rationale Correctness** The initial evaluation centred on the accuracy of rationales. Annotators evaluated the rationales based on two primary criteria: (1) Correctness of matched key elements: Evaluating whether the rationale correctly identifies key elements mentioned by the student's answer. (2) Faithfulness of rubric application: Reviewing if the used rubric corresponds appropriately with the score assigned to the student's answer and the elements identified in the rationale.

**Preference on Rationale** The subsequent evaluation was tailored towards annotators' preferences concerning the rationales. Annotators were shown rationales generated by AERA and ChatGPT in a randomized order. Their task was to choose the rationale they deemed superior, basing their decision on factors such as accuracy and informativeness. The chosen rationale was gauged on its ability to

---

aptly convey pertinent details and its efficacy in substantiating the student answer assessment.

**Overall Analysis** The left segment of Figure 3 indicates that AERA-generated rationales considerably surpass ChatGPT-generated ones across both evaluation criteria. Given the inherent challenge for language models to pinpoint key elements resonating with student answers, it's noticeable that scores tend to be lower for the correctness of matched key elements compared to the rubric application's faithfulness for both models.

The right segment of Figure 3 underscores a marked inclination among annotators towards AERA-generated rationales over ChatGPT's. Despite LLMs sometimes offering more expansive explanations due to their in-context learning prowess, they frequently underperform in accurately gauging student answers relative to our refined model, leading to a diminished preference rating.

In summary, the compact language model distilled using the AERA framework not only outperforms ChatGPT in student answer assessment but also produces more precise rationales, despite its significantly smaller size compared to ChatGPT.

### 4.4 Case Studies on Refinement Strategies

---

**#1**: In order to replicate this procedures, you would need to
**Original Label**: 3     **High Confident Prediction**: 0
**Rationale**: This response is incomplete and does not provide any relevant information.

---

**#2**: In conclusion trial any
**Original Label**: 2     **High Confident Prediction**: 0
**Rationale**: The student's response is incomplete and does not provide a valid conclusion or any ways to improve the experimental design and/or the validity of the results.

---

**#1**: In order to replicate this experiment, you would need to know: 1. how they got the mass of the four different samples 2. A list of constants 3. You would have to know how much of a sample you would place into the container of vinegar and if it's the same for all four materials.
**Original Label**: 0     **High Confident Prediction**: 2
**Rationale**: This response describes two additional pieces of information that would be needed to accurately replicate the experiment: "how they got the mass of the four different samples" and "you would have to know how much of a sample you would place into the container of vinegar and if it's the same for all four materials." The response also mentions a list of constants, but it is not clear what specific additional information is being requested.

---

Table 2: Examples of Incorrectly Labeled Data.

**Identification of Incorrectly Labeled Data** As shown in Table 2, we discover that highly confident incorrect predictions by ChatGPT may actually be correct using the method outlined in §3.2, suggest-

---

[7]For a comprehensive evaluation setup, data statistics, IAA scores, and various breakdown evaluation results, refer to §B.5

| |
|---|
| **#1**: *The students data needed to include how much of the solution was poored. They also needed to find out if the rocks weighed the same when they started.*
**Original**: 1 point; This response describes one piece of relevant information that would be needed to accurately replicate the experiment: "how much of the solution was poured."
**Refined**: *0 points;* This response describes little or no accurate or relevant information from the acid rain investigation. |
| **#5**: *There four major steps of protien synthesis. The first is the mRNA going to the rRNA. Then the rRNA takes the information from the mRNA to the tRNA. Next the tRNA will take the information it has and translate it. Lastly, the tRNA will take the information and make a protein.*
**Original**: 2 points; The student answer matches two key elements, "...mRNA going to the rRNA..." and "...tRNA will take the information and make a protein...". However, the other two steps are not described accurately or comprehensibly.
**Refined**: *1 point;* The student answer matches only one key element, "...mRNA going to the rRNA..." |
| **#6**: *1.-Osmosis is how water gets diffused thru the membrane. 2.-Active Transport is where an enzyme opens the cell membrane for an object to come in, and extra energy is needed to assist the object into the cell.3.-Passive Transport is where an enzyme opens the cell, but the object doesn't need the extra energy to come in.*
**Original**: 2 points; This student answer matches two key elements, "Osmosis... how water gets diffused thru the membrane" and "Passive Transport... enzyme opens the cell, but the object doesn't need the extra energy to come in". However, the explanation for "Active Transport..." is not entirely accurate, as it mentions an enzyme opening the membrane, which is not correct. Also, the explanation doesn't match the entire description of active transport.
**Refined**: *3 points;* This student answer matches three key elements, "Osmosis... how water gets diffused", "Active transport... enzyme opens the cell membrane for an object to come in, and extra energy is needed" and "Passive transport... enzyme opens the cell, but the object doesn't need the extra energy to come in". |

Table 3: Improved Rationale Examples Using the Rationale Refinement Strategy.

ing that the data may be noisy or mislabelled. For example, in the first two cases, student answers are incomplete, possibly due to data corruption. The discrepancy between the human labels and the actual answers highlights the clear mismatch in the original dataset. Besides, we also identify instances that may have been annotated with incorrect lower scores. For instance, the last example in the table clearly covers two key elements based on the rubric (highlighted in orange), but the original score given is 0 point. Such mislabeled data could be difficult to detect without manual examination in a text classification setup. The above discoveries from the dataset, which have not been highlighted in previous research, serve as a validation of our concern regarding the presence of inconsistent marking standards in large-scale student answer assessments. Our approach provides a feasible solution to automatically identifying label inconsistency or data corruptions in a human-annotated dataset.

**Rationale Refinement**    As shown in Table 3, we demonstrate that by providing the *correct assessment points* in the prompt, ChatGPT is able to improve its generated rationale better aligned with the score provided. For example, in the first case, an incorrect key element was initially identified. However, after the correct score is provided to ChatGPT, the model is able to correctly trace back the applicable rubric and thus decides that no key elements were mentioned in the text. We discovered that the original incorrect identification might have been

influenced by the presence of "*Other acceptable responses*" stated in the key elements. Determining which part of the response falls into the "*acceptable*" category can be challenging for ChatGPT. The other two examples demonstrated common mistakes in human annotations that occurred in the dataset. In these two cases, ChatGPT might have misinterpreted some student descriptions, but the refinement step is able to rectify the mismatches in key elements. However, this strategy cannot be applied if the data contains wrongly labelled instances, as ChatGPT will be forced to generate rationales that may not make sense. Given the above observations, we urge the need for future development of student answer assessment datasets to provide enough examples for key elements. This could help mitigate ambiguous definitions and provide clearer guidelines for key elements, thereby reducing confusion and improving the consistency of the student answer assessment process.[8]

## 5   Conclusion

In this paper, we proposed a framework called **AERA**, which leverages the in-context learning and reasoning capabilities of ChatGPT for rationale generation in student answer assessment. Our experimental results suggest that although ChatGPT is able to generate free-form rationales with natural language instructions, the example instructed prompt strategy achieves the best performance. We

---

[8]We provide further experimental details and comprehensive ablation studies in §B.

further demonstrate **AERA** can effectively distil a smaller language model for efficient rationalization on automated student answer assessment tasks, without the need for additional human annotation on rationales. Extensive experiments and human evaluation results have validated the efficacy of the refinement module, and our distilled language model can outperform the teacher model in grading while providing reliable rationales. Our approach presents a cost-effective and efficient method for explainable automated student answer assessment.

## Limitations

This study has several limitations. First, there may be variations in the designs of prompt templates among individuals, and the manual prompt performance can differ across different datasets. Moreover, due to the extensive search space involved in generating automated prompt text, the auto prompt approach cannot be adequately tested with our current computational resources. Second, although appropriate training has been provided for the annotators, the lack of background in exam assessment among the human evaluation annotators may have some impact on the quality of the evaluations. Lastly, we identified a trade-off between interpretability and assessment performance. Given the variations in base models and structures, bridging this gap remains challenging at present.

## Ethics Statement

The dataset utilized in this study is an open-source collection of anonymous student responses, and does not contain any sensitive or identifiable information. Although we have not identified any harmful outputs from ChatGPT in our study, it is worth noting that previous research has observed instances where ChatGPT produced unexpected results. We encourage other researchers to utilize this framework to scrutinize the output generated from specific prompts in ChatGPT that may have the potential to generate harmful information.

## Acknowledgements

This work was supported in part by the UK Engineering and Physical Sciences Research Council (grant no. EP/T017112/2, EP/V048597/1, EP/X019063/1). JL is funded by a PhD scholarship provided by AQA. YH is supported by a Turing AI Fellowship funded by the UK Research and Innovation (grant no. EP/V020579/2).

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

## A    Further Framework Details

### A.1    Tabular Data Transformation

Some questions in our dataset contain tabular data, which poses a challenge for smaller language models in terms of inputting and understanding structured data. To address this issue, as shown in Figure A1, we leverage ChatGPT's table-understanding capability to create table descriptions from the tabular data and verify the description correctness by having ChatGPT generate a table based on the description[9]. Notably, we found that all the tabular data in our dataset could be accurately reconstructed based on the description generated by ChatGPT. Consequently, we replaced all the tabular data from the question part of our prompts with the corresponding generated descriptions.

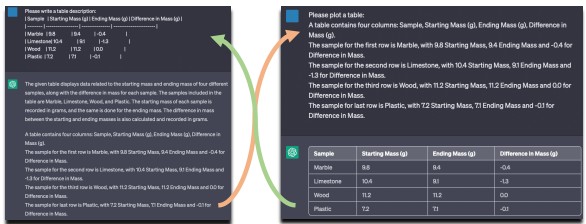

Figure A1: Demonstration of using ChatGPT for tabular data and table description transformation.

## B    Further Experimental Details and Discussions

### B.1    Experimental Setup

**Dataset**    In this paper, we employ the Hewlett Foundation: Short Answer Scoring (ASAP-SAS) dataset[10]. This dataset encompasses over 23,000 short answer responses from students in grades 7 to 10, including ten questions spanning subjects such as Science, Biology, English, and Art. Expert human raters have manually scored each response on a scale of 0-2 or 0-3, based on predefined rubrics. Instead of focusing on assessment on the grammatical or writing side of the student responses, we are more interested in response assessment on STEM-related questions. Therefore, we only selected four subsets (#1, #2, #5 and #6) relating to Science and Biology from the ASAP-SAS datasets. We didn't include other subsets since they are either focused on English and Art or contain multi-modal data (e.g. Graphs) in the question that is difficult to be fed into language models. As the original dataset

only provides the training and test sets, we created a development set by partitioning the training set in an 8:2 ratio. The detailed train, development, and test splits are shown in Table A1.

| Subset | **#1** | **#2** | **#5** | **#6** |
|---|---|---|---|---|
| # Train | 1,338 | 1,023 | 1,436 | 1,438 |
| # Dev | 334 | 255 | 359 | 359 |
| # Test | 557 | 426 | 598 | 599 |

Table A1: Dataset statistics.

**Quadratic Weighted Kappa Implementation**
Quadratic Weighted Kappa, a widely used metric in evaluating the agreement between two raters in student response assessment, is defined as:

$$\kappa = 1 - \frac{\sum_{i=1}^{k} \sum_{j=1}^{k} w_{ij} O_{ij}}{\sum_{i=1}^{k} \sum_{j=1}^{k} w_{ij} E_{ij}} \quad (1)$$

where $k$ is the score set, $w$ is the weighted matrix, calculates as: $w_{i,j} = \frac{(i-j)^2}{(k-1)^2}$. $O$ is a $k \times k$ histogram matrix and $E$ being the $k \times k$ expected value matrix.

**Hyperparameter Settings**    We utilized the OpenAI API with the `gpt-3.5-turbo` model version 23 Mar 2023 for the generation of *Simple/Complex/Example instruction*-based rationales. Default parameters were retained, with the temperature parameter set to 1.0. For our fine-tuning experiments, we deployed NVIDIA A100 80G graphics cards. The AERA fine-tuning procedure adopted the `Long-t5-tglobal-large` as the foundational model. Training for the rationale generation (RG) task was executed with a batch size of 8 over 30 epochs, while the text classification (TC) task used a batch size of 16 across the same number of epochs. We selected learning rates of 1e-5 for the TC task and 1e-4 for the RG task, implementing a weight decay of 0.01. To ensure robust performance metrics, each configuration was executed thrice for RG and five times for TC, using random seeds of 210, 102, 231, 314, and 146.

**Model Implementation**    We utilized the HuggingFace Transformer library[11] for the implementation of models such as BERT (Devlin et al., 2019), Longformer (Beltagy et al., 2020), and LongT5[12] (Guo et al., 2022).

---

[9]Obtained with ChatGPT version 13th Feb 2023.
[10]https://kaggle.com/competitions/asap-sas

[11]HuggingFace Transformer
[12]Given the extensive nature of the content within questions, key elements, and rubrics, the combined length with student responses typically exceeds 1,024 tokens. Consequently, our experiments employ models specifically designed to manage inputs from longer documents.

## B.2 Faithfulness of ChatGPT-Generated Rationales w.r.t its Predicted Scores

To the best of our knowledge, there is no established automated evaluation method for assessing the quality of ChatGPT-generated rationales. We proposed to design a proxy check to verify the faithfulness of the ChatGPT-generated rationale with respect to its predicted student answer assessment scores, which can be represented as $R \rightarrow Y$. We gathered the outputs produced by ChatGPT on our dataset and fine-tuned a text classifier to predict the score $\hat{y}_i$ using the generated rationale $\hat{r}_i$ as input. In this process, we did not perform any filtering. That is, some of the ChatGPT-predicted answer scores may be wrong. Our purpose is to establish a proxy check if the ChatGPT-generated rationales are indeed faithful explanations of its predicted answer scores. As shown in Table A2, we observe a strong correlation between the ChatGPT-generated rationales and its predicted corresponding scores across all four datasets. This finding suggests that the ChatGPT-generated rationales could be considered as somewhat faithful explanations of its predicted assessment scores.

|     | #1    | #2    | #5    | #6    |
|-----|-------|-------|-------|-------|
| Acc | 98.69 | 87.08 | 98.15 | 93.10 |
| F1  | 98.77 | 83.47 | 97.18 | 95.47 |
| QWK | 99.07 | 90.48 | 97.87 | 93.36 |

Table A2: Predictive performance on score classification output by ChatGPT using its generated rationales.

## B.3 Simulatability of ChatGPT-Generated Rationales w.r.t its Predicted Scores

Wiegreffe et al. (2021) proposed a rationale quality evaluation method based on the association between generated rationale and the predicted label to evaluate the free-text rationale quality. Simulatability, instead of relying on the word-level overlap, assesses the ability of a generated rationale to predict the label by measuring the difference in task performance when the rationale is provided as input compared to when it is absent:

$$acc(IR \rightarrow O) - acc(I \rightarrow O) \qquad (2)$$

We conducted an experiment to evaluate the simulatability of rationales generated by ChatGPT, as detailed in Table A3. In this context, $XR \rightarrow Y$ denotes a generative classification setting fine-tuned on the Long T5 model. It takes into account questions, key elements, rubrics, student answers, and ChatGPT-generated rationales (using the Example Instruction template) as input, and outputs ChatGPT-predicted scores. Conversely, $X \rightarrow Y$ is tuned under the same classification setting but omits the rationale from the input.

Contrary to the consistency findings from (Wiegreffe et al., 2021), where results trended toward 0, we noted positive disparities between $acc(XR \rightarrow Y)$ and $acc(X \rightarrow Y)$, as evident in the table's final row. This implies that rationales generated by ChatGPT, utilizing the *Example Instruction* template, enhance label prediction, especially for datasets #1, #2, and #5. While the accuracy difference for dataset #6 is less than 0, there's a marked improvement in F1 and QWK metrics. This suggests that incorporating rationales into the input bolsters class sensitivity and aligns more closely with gold label scores.

In summary, across all datasets, the performance uptick indicates that ChatGPT-produced rationales exhibit commendable quality in simulatability tests. However, dataset #6's outcomes hint that solely focusing on accuracy for evaluations might not be ideal for tasks with nuanced class sensitivity, such as student answer assessment.

## B.4 Results by Fine-Tuning Long T5 on Filtered ChatGPT Outputs

In Table A4, we present the statistics of the training set after filtering out instances where ChatGPT predicts wrong answer scores. We observe that when using the *Simple Instruction*, ChatGPT predicts correct answer scores for less than half of the instances. However, with the *Complex Instruction*, there is a notable increase in the number of instances where ChatGPT outputs correct answer scores. Interestingly, the *Example Instruction* does not yield improvements for dataset #1, #5, and #6. But it enables ChatGPT to predict more correct answer scores for the dataset #2.

In Table A5, we show the results by fine-tuning Long T5 on the filtered ChatGPT outputs. Consistent with ChatGPT's inference performance, the fine-tuned Long T5 also exhibits performance improvement when trained on the filtered ChatGPT outputs produced using *Complex* or *Example Insutrctions* compared to *Simple Instruction*. Interestingly, although the amount of data is reduced for subsets #1, #5, and #6 when using the *Example Instruction* compared to *Complex Instruction* as

| Dataset (Subject) | #1 (Science) | | | #2 (Science) | | | #5 (Biology) | | | #6 (Biology) | | |
|---|---|---|---|---|---|---|---|---|---|---|---|---|
| Method/Model | Acc | F1 | QWK | Acc | F1 | QWK | Acc | F1 | QWK | Acc | F1 | QWK |
| $X \to Y$ | 69.96 | 70.84 | 82.09 | 56.57 | 53.12 | 57.83 | 85.62 | 58.94 | 79.53 | 89.20 | 62.86 | 83.19 |
| $XR \to Y$ | 80.91 | 77.53 | 85.70 | 82.39 | 80.52 | 87.61 | 87.34 | 82.56 | 88.89 | 88.48 | 76.83 | 89.58 |
| $acc(XR \to Y) - acc(X \to Y)$ | +10.95 | - | - | +25.82 | - | - | +1.72 | - | - | -0.72 | - | - |

Table A3: Analysis on ChatGPT-generated Rationales' Simulatability.

| Subset | #1 | #2 | #5 | #6 |
|---|---|---|---|---|
| # Train | 1,338 | 1,023 | 1,436 | 1,438 |
| Simple Instruction | 627 | 412 | 761 | 692 |
| Complex Instruction | 695 | 407 | 1,051 | 1,016 |
| Example Instruction | 689 | 477 | 968 | 987 |

Table A4: Statistics of the training set after filtering out incorrect ChatGPT-predicted answer scores.

shown in Table A4, the overall performance is the best for the fine-tuned Long T5 models. We have conducted error analysis on the ChatGPT-generated outputs and found that the hallucination problem could be significantly reduced by providing demonstration examples in the *Example Instruction* (More discussions can be found in §B.6). For this reason, we decided to use the *Example Instruction* in all our subsequent experiments.

## B.5 Human Evaluation Details

In this section, we provide further details and settings on our human evaluation experiments.

### B.5.1 Evaluation Setup

**Data Selection**  We randomly selected 10% of instances from the run with the highest QWK. Among the sampled data, we further selected 20% for the purpose of calculating the Inter-Annotator Agreement (IAA) score. The detailed statistics of the total sampled data are shown in Table A6.

**Annotator**  Two annotators are selected for the evaluation process. Both evaluators are PhD students with computer science backgrounds and have received training on the evaluation schema and the use of the annotation platform. Each assigned task took about 5 hours to complete, and the annotators were paid fairly at a rate of $21.83/hour.

**Evaluation Platform**  As shown in Figure A2, our evaluation is built with Docanno[13]. The labels are designed to indicate whether an option is considered correct or incorrect based on its selection or non-selection.

[13] https://github.com/doccano/doccano

### B.5.2 Human Evaluation Results

**IAA Results**  We use Cohen Kappa for IAA analysis.

$$\kappa = 1 - \frac{1 - P_o}{1 - P_e}$$

where $P_o$ is the relative observed agreement among raters (identical to accuracy), and $P_e$ is the hypothetical probability of chance agreement, using the observed data to calculate the probabilities of each observer randomly seeing each category. Our IAA results in Table A7 show that the annotators exhibited moderate agreement on the correctness of key elements and the faithfulness of rubric, while they fairly agreed on the preference of rationales.

**More Detailed Evaluation Results**  We present a breakdown of evaluation results for both human evaluation tasks, showing the percentage of correctness selections by each annotator in Table A8, and by each subset in Table A9.

### B.6 Analysis of ChatGPT Hallucinations

In this section, we discuss various hallucination cases observed in the ChatGPT-generated rationales under the zero-shot setting, i.e., using either *Simple Instruction* or *Complex Instruction* as described in §3.1 and without supplying any demonstration examples. Table A10 demonstrates cases of inconsistent and inaccurate assessments, which can be grouped into five types: **(1) Incorrect scoring scale**. Despite providing a clear 0-3 integer score rubric, ChatGPT occasionally generates rationales that include incorrect score caps, such as 5 or 12, or even fractional score scales possibly stem from its knowledge base. **(2) Inconsistent assessment**. Some rationales display completely contradictory scores in two different places in the rationale text. **(3) Uncertain score prediction**. In some cases, ChatGPT may ignore the marking rubric and outputs uncertain scores such as '*1-2 points*'. **(4) Factual mistake**. We observed instances where the matched key elements identified in the generated rationales were never mentioned by the student's answer or included in the key answer elements. **(5)**

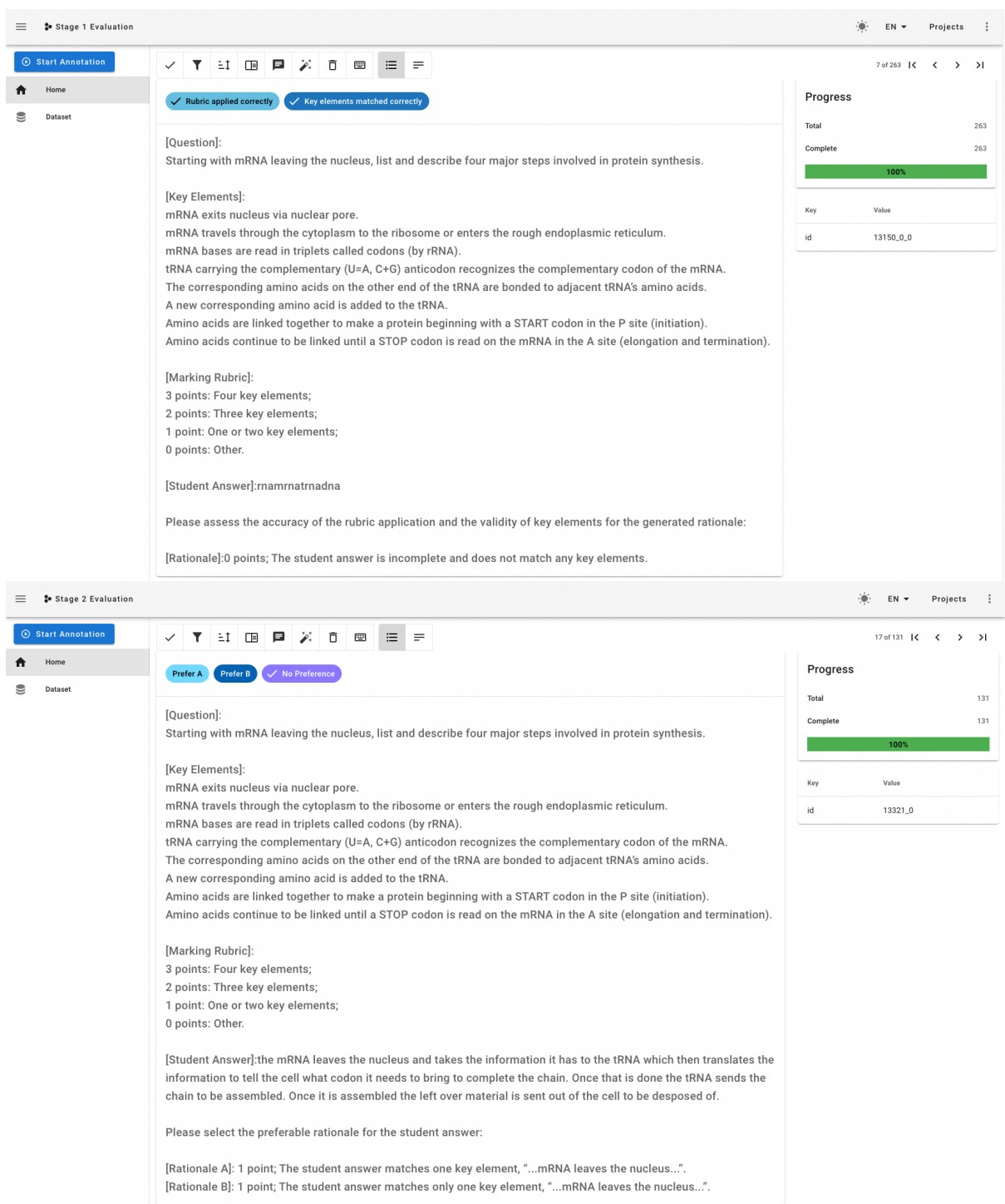

Figure A2: Screenshots of the annotation platform for both human evaluation tasks.

| Dataset (Subject) | #1 (Science) | | | #2 (Science) | | | #5 (Biology) | | | #6 (Biology) | | | Overall | | |
|---|---|---|---|---|---|---|---|---|---|---|---|---|---|---|---|
| Method/Model | Acc | F1 | QWK | Acc | F1 | QWK | Acc | F1 | QWK | Acc | F1 | QWK | Acc | F1 | QWK |
| Simple Instruction | 43.39 | 32.39 | 40.01 | 23.71 | 9.97 | 0.69 | 68.56 | 32.82 | 45.29 | 79.69 | 37.14 | 64.95 | 53.84 | 28.08 | 37.74 |
| Complex Instruction | 47.16 | 38.36 | 54.48 | 40.61 | 29.3 | **38.3** | 79.21 | 42.14 | 61.63 | 85.70 | 43.26 | 67.73 | 63.17 | 38.27 | 55.54 |
| Example Instruction | 56.79 | 55.95 | **69.96** | 35.60 | 24.02 | 23.94 | 78.76 | 48.79 | **71.83** | 84.19 | 54.93 | **79.17** | 63.84 | 45.92 | **61.23** |

Table A5: Evaluating the performance of Long T5 models that have been fine-tuned using rationales generated by ChatGPT prompt with other templates.

| Subset | #1 | #2 | #5 | #6 | #all |
|---|---|---|---|---|---|
| 10% Sampled | 56 | 43 | 60 | 60 | 219 |
| Duplicate for IAA | 11 | 9 | 12 | 12 | 44 |
| Total | 67 | 52 | 72 | 72 | 263 |
| Instances for Rationale Correctness | | | | | 526 |
| Instances for Rationale Preference | | | | | 263 |

Table A6: The statistics of the sampled data for human evaluation.

| Tasks | IAA Score |
|---|---|
| Correctness of Key Elements | 0.4579 |
| Faithfulness of Rubric | 0.5056 |
| Rationale Preference | 0.3276 |

Table A7: Inter-Annotator Agreement results.

| | Ann 1 | Ann 2 | Total |
|---|---|---|---|
| Human Evaluation on Rationale Correctness | | | |
| Key Elements on AERA | 0.86 | 0.80 | **0.83** |
| Rubric on AERA | 0.96 | 0.92 | **0.94** |
| Key Elements on ChatGPT | 0.60 | 0.38 | 0.52 |
| Rubric on ChatGPT | 0.86 | 0.87 | 0.86 |
| Human Evaluation on Rationale Preference | | | |
| Prefer AERA | 0.57 | 0.50 | **0.54** |
| Prefer ChatGPT | 0.24 | 0.22 | 0.23 |
| No Preference | 0.17 | 0.28 | 0.23 |

Table A8: A breakdown of evaluation results by each annotator.

| | #1 | #2 | #5 | #6 |
|---|---|---|---|---|
| Human Evaluation on Rationale Correctness | | | | |
| Key Elements on AERA | 0.92 | 0.72 | 0.90 | 0.73 |
| Rubric on AERA | 1.00 | 0.78 | 1.00 | 0.93 |
| Key Elements on ChatGPT | 0.48 | 0.51 | 0.53 | 0.54 |
| Rubric on ChatGPT | 0.91 | 0.85 | 0.96 | 0.73 |
| Human Evaluation on Rationale Preference | | | | |
| Prefer AERA | 0.56 | 0.48 | 0.56 | 0.53 |
| Prefer ChatGPT | 0.16 | 0.29 | 0.24 | 0.26 |
| No Preference | 0.27 | 0.23 | 0.21 | 0.21 |

Table A9: A breakdown of evaluation results by each subset.

**Vague rationale**. We observed that zero-shot generated rationales often provide vague or irrelevant explanations for student response, which may not be helpful for feedback and could be difficult to understand.

In contrast, using the *Example Instruction* prompt by supplying some demonstration examples guides ChatGPT to follow a structured format for rationale generation and answer scoring. Moreover, instructions that are oriented towards examples help ChatGPT to rely less on its knowledge base and instead utilise information from the provided resources such as key answer elements and marking rubric. Our analysis reveals that the hallucination problem can be partly alleviated by using the *Example Instruction* prompt with demonstration examples. Consequently, we have chosen the *Example Instruction* prompt as our primary rationale generation method.

## B.7 Example Rationales Generated using AERA vs. ChatGPT

Table A13 shows example rationales generated using the student model, Long T5, in comparison with those generated by the teacher model, ChatGPT. We observe that both the Long T5- and ChatGPT-generated results follow the same structured format

as demonstrated in the examples provided in the prompt, that a score is given first, followed by a rationale explaining the scoring decision.

The refinement of the training data, which involved cleaning and correcting some inaccurately generated rationales by providing the actual answer scores as input to ChatGPT, has led to a stronger correlation between Long T5-generated rationales and the predicted scores. On the contrary, the ChatGPT-generated results for #1, #2, and #6 exhibit minor discrepancies due to over-matching or under-matching certain key elements.

We also noticed a small number of mistakes in the Long T5-generated results, primarily attributable to the students' vague descriptions, making it difficult for the language model to compare the answers with the key elements. Additionally,

**Incorrect scoring scale:**
... answer should receive 1 point out of 5.
... answer should receive 1.5 points out of 3.
... Overall, this student answer receives a score of 2 out of 12 (0+0+1+1) as the answer does not accurately and completely ...

**Inconsistent assessment:**
Score: 1 point This student answer ... Therefore, the answer is not relevant to the question and should receive a score of 0 points.

**Uncertain score prediction:**
... Therefore, this answer would receive a score of 1-2 points out of 3.

**Factual mistake:**
... this Student answer includes three of the key elements: selective permeability, passive transport, and facilitated diffusion

**Vague rationale:**
... the answer demonstrates some understanding of protein synthesis but is missing several key elements and contains some inaccuracies.

Table A10: ChatGPT hallucination examples from the rationales generated using either *Simple* or *Complex* Instruction under the zero-shot setting.

some questions include rubrics such as "*other acceptable responses*", which are particularly challenging for language models to assess, given their lack of domain-specific background knowledge.

In summary, our distilled Long T5 model demonstrates a strong capability to assess student responses and generate accurate rationales. Despite the occasional errors and challenges posed by vague student answers and certain rubrics, the model's overall performance is promising for applications in educational settings.

## B.8 Explore the Influence of Number of Demonstration Examples

In this section, we have performed an ablation study on the influence of test performance by the number of demonstrations provided to ChatGPT[14]. In this experiment, we gradually reduced the number of demonstration examples included in the prompt to find out the influence on the performance. As we present on the #6 in Table A11, aligned with observations reported in prior work (Brown et al.,

[14]ChatGPT version 3 Aug 2023.

2020), the test performance achieves the highest with all the demonstration examples included.

## B.9 Investigate the Generalizability of AERA

We wanted to demonstrate that our approach is applicable in a wide range of scenarios. To do this, we conducted an ablation study called "leave one out", training our framework on three subsets and testing it on the left subset. The results, as shown in Table A12, indicate that our framework can not only evaluate student answers based on the trained question, key elements and rubric; but also generalize well beyond to unseen datasets.

## B.10 Rationale Generation from Other LLMs

This section presents an example from the #5 to demonstrate that our prompting strategy is still effective for models other than ChatGPT, such as Bard or FlanT5. During the experiment design phase, we primarily focused on ChatGPT due to its robust capabilities and cost-effectiveness. The largest open-sourced model We experimented with was the LLaMA-2 70B. However, as shown in Table A14, the model struggled to produce coherent rationales and often repeated the marking rubrics in its response.

| Demo amount | Scores | Acc | F1 | QWK |
|---|---|---|---|---|
| 5 | 0,1,2,3,3 | **79.97** | **54.66** | **71.48** |
| 4 | 0,1,2,3 | 77.96 | 40.75 | 68.63 |
| 3 | 0,1,2 | 74.62 | 48.75 | 66.24 |
| 2 | 0,1 | 72.62 | 42.82 | 54.90 |
| 1 | 0 | 67.45 | 39.30 | 57.26 |

Table A11: Ablation study on the number of demonstrations on #6.

| Train Sets | Test Sets | Acc | F1 | QWK |
|---|---|---|---|---|
| #2, #5, #6 | #1 | 57.99 | 56.05 | 71.20 |
| #1, #5, #6 | #2 | 44.84 | 40.34 | 58.42 |
| #1, #2, #6 | #5 | 84.95 | 39.49 | 63.02 |
| #1, #2, #5 | #6 | 85.14 | 54.10 | 75.21 |

Table A12: "leave one out" experiment for method generalizability.

**#1 Student Answer:** To replicate the group's experiment, the procedure would have to state how much vinegar was poured into the containers. Also, they should specify what kinds of containers they were using as this could affect the results. In the procedure, they also failed to tell what 4 samples were used in the experiment, a key variable to it's success. **Score: 3**

**Long T5:** 3 points; This response describes three additional pieces of information that would be needed to accurately replicate the experiment: ,"how much vinegar was poured into the containers...what kinds of containers they were using...and what 4 samples were used in the experiment.,"

**ChatGPT:** 2 points; This response describes two additional pieces of information that would be needed to accurately replicate the experiment: ,"how much vinegar was poured into the containers," and ,"what 4 samples were used in the experiment.," It also hints at a potential third piece of information needed: ,"what kinds of containers were used.,"

---

**#2 Student Answer:** a. Based on the data, the stretchiest polymer plastic of the four is plastic type B, because in the both trials, it stretched the most (22 and 23 mm). b. 1.) Add more trials to increase accuracy of the results. 2.) Repeat the experiment with heavier weights to see if it affects the results. **Score: 2**

**Long T5:** 2 points; The student provides an acceptable conclusion based on the data: ,"...the stretchiest polymer plastic of the four is plastic type B....," One correct way to improve the experimental design and/or the validity of the results is given: ,"Add more trials to increase accuracy of the results.," The second improvement given, ,"...trying the experiment with heavier weights to see if it affects the results,," is too vague and does not receive credit.

**ChatGPT:** 3 points; The student provides an acceptable conclusion based on the data: ,"...the stretchiest polymer plastic of the four is plastic type B...," Two correct ways to improve the experimental design and/or the validity of the results are given: ,"1.) Add more trials to increase accuracy of the results....," and ,"2.) Repeat the experiment with heavier weights to see if it affects the results.,"

---

**#5 Student Answer:** tRNA will transfer it out. **Score: 0**

**Long T5:** 0 points; The student answer does not match any key elements given.

**ChatGPT:** 0 points; The student answer does not provide any coherent or relevant information on the steps involved in protein synthesis.

---

**#6 Student Answer:** Osmosis, Is the movement of water across the membrain. Endocitosis, is the movement of things in to the cell. Exocitosis, is the movement of things out of the cell. **Score: 1**

**Long T5:** 1 point; This student answer only matches one key element, "Osmosis... movement of water". The other two concepts are incorrect or incomplete.

**ChatGPT:** 2 points; This student answer matches two key elements, "Osmosis... movement of water across the membrane" and "Endocytosis... movement of things into the cell" but didn't include an explanation for "Exocytosis".

Table A13: Examples of AERA generated rationales compared with ChatGPT results.

## C    Prompt Details

In this section, we provide the full detail of the question, key elements and rubric we used in the prompt for each dataset. We highlighted table descriptions generated via Tabular Data Transformation mentioned in §A.1 in orange.

### C.1    Subset #1

[Question]:
A group of students wrote the following procedure for their investigation.
Procedure:
1.Determine the mass of four different samples.
2.Pour vinegar in each of four separate, but identical, containers.
3.Place a sample of one material into one container and label. Repeat with remaining samples, placing a single sample into a single container.
4.After 24 hours, remove the samples from the containers and rinse each sample with distilled water.
5.Allow the samples to sit and dry for 30 minutes.
6.Determine the mass of each sample.
The students's data are recorded in the table below. A table contains four columns: Sample, Starting Mass (g), Ending Mass (g), Difference in Mass (g). The sample for the first row is Marble, with 9.8

***Given this student answer from #5 with the Example Instruction prompt template:***
"1. tRNA tranfers it outside the nucleus to be matched with a ribosome2. The ribosome latches on to an amino acid.3. The codons match up4. It creates protein."

***ChatGPT***: 1 point; The student answer only matches one key element, "...tRNA transfers it outside the nucleus to be matched with a ribosome...".

***GPT-4***: 1 point; The student answer matches two key elements, "The codons match up" and "It creates protein".

***Google Bard:*** The student answer matches only two key elements, 'tRNA tranfers it outside the nucleus to be matched with a ribosome' and 'The codons match up', so the score is 1 point. Here are the four major steps involved in protein synthesis, starting with mRNA leaving the nucleus: Initiation: mRNA leaves the nucleus and binds to a ribosome in the cytoplasm. The ribosome then scans the mRNA for a start codon (AUG). Elongation: A tRNA molecule with an amino acid attached to it binds to the ribosome at the A site. The anticodon of the tRNA matches the start codon on the mRNA. The amino acid is then added to the growing polypeptide chain. The ribosome then moves one codon down the mRNA. Termination: The ribosome continues to read the mRNA until it reaches a stop codon. A release factor binds to the stop codon and the polypeptide chain is released from the ribosome. Folding and modification: The newly synthesized polypeptide chain is folded into its correct shape. It may also be modified by the addition of chemical groups. The student answer does not mention the initiation step, which is a critical step in protein synthesis. Therefore, the answer is not complete and the score is 1 point.

***Flan T5 XXL***: *(Exceeded max input sequence length)* 1 point; The student answer matches one key element, "...tRNA tranfers it outside the nucleus to be matched with a ribosome2. The ribosome latches on to an amino acid3. The codons match up4. It creates protein.".

***LLaMA 2 70B Chat HF***: Sure, here's the scoring rubric for the student answers: [Student Answer]: 1. tRNA tranfers it outside the nucleus to be matched with a ribosome2. The ribosome latches on to an amino acid.3. The codons match up4. It creates protein. [Rationale and Score]: The student answer matches two key elements: "tRNA tranfers it outside the nucleus to be matched with a ribosome" (step 1) and "It creates protein" (step 4).* However, the student answer does not mention the other two key elements: "mRNA bases are read in triplets called codons (by rRNA)" (step 2) and "Amino acids are linked together to make a protein beginning with a START codon in the P site (initiation)" (step 3). Score: 2 points Here's the corrected scoring rubric:* 4 points: Four key elements;* 3 points: Three key elements;* 2 points: Two key elements;* 1 point: One key element;* 0 points: Other.

Table A14: Example output for answer assessment and rationale generation from other popular LLMs.

Starting Mass, 9.4 Ending Mass and -0.4 for Difference in Mass.

The sample for the second row is Limestone, with 10.4 Starting Mass, 9.1 Ending Mass and -1.3 for Difference in Mass.

The sample for the third row is Wood, with 11.2 Starting Mass, 11.2 Ending Mass and 0.0 for Difference in Mass.

The sample for last row is Plastic, with 7.2 Starting Mass, 7.1 Ending Mass and -0.1 for Difference in Mass.

After reading the group's procedure, describe what additional information you would need in order to replicate the experiment.

Make sure to include at least three pieces of information.

[Key Elements]:
Needed Information:
You need to know how much vinegar was used in each container.
You need to know what type of vinegar was used in each container.
You need to know what materials to test.
You need to know what size/surface area of materials should be used.
You need to know how long each sample was rinsed in distilled water.
You need to know what drying method to use.
You need to know what size/type of container to use.
Other acceptable responses.

[Rubric]:
3 points: The response describes three additional pieces of information that would be needed to accurately replicate the experiment;
2 points: The response describes two additional pieces of information that would be needed to accurately replicate the experiment;
1 point: The response describes one additional piece of information that would be needed to accurately replicate the experiment;
0 point: The response describes little or no accurate

or relevant information from the acid rain investigation.

### C.2 Subset #2

[Question]:
A student performed the following investigation to test four different polymer plastics for stretchability.
Procedure:
1. Take a sample of one type of plastic, and measure its length.
2. Tape the top edge of the plastic sample to a table so that it is hanging freely down the side of the table.
3. Attach a clamp to the bottom edge of the plastic sample.
4. Add weights to the clamp and allow them to hang for five minutes.
5. Remove the weights and clamp, and measure the length of the plastic types.
6. Repeat the procedure exactly for the remaining three plastic samples.
7. Perform a second trial (T2) exactly like the first trial (T1).
The student recorded the following data from the investigation.
The table shows the amount of stretch (in millimeters) for four different types of plastic, labeled as A, B, C, and D, when subjected to two different stretching forces, labeled as T1 and T2.
For plastic type A, it stretched 10mm under T1 and 12mm under T2.
For plastic type B, it stretched 22mm under T1 and 23mm under T2.
For plastic type C, it stretched 14mm under T1 and 13mm under T2.
Lastly, for plastic type D, it stretched 20mm under both T1 and T2.
a. Draw a conclusion based on the student's data.
b. Describe two ways the student could have improved the experimental design and/or validity of the results.

[Key Elements]:
Conclusions:
Plastic sample B has more stretchability than the other polymer plastics.
Plastic sample A has the least amount of stretchability compared to the other polymer plastics.
Not all polymer plastics have the same stretchability.

Different polymer plastics have different stretchability (and are therefore suited for different applications).
A reasonable conclusion cannot be drawn due to procedural errors.
Other reasonable conclusions
Experimental Design Improvements:
Provide the before and after measurements for length (Did the samples all start out the same size?).
Make sure the samples are all of the same thickness.
Variations in thickness could have caused variations in stretchability.
Perform additional trials.
Some of the samples have similar stretchability (A and C, B and D).
Two trials may not be enough to conclusively state that one is more stretchable than the other.
Indicate how many weights were added to the clamps (Was it the same number for each sample?).
Other acceptable responses

[Rubric]:
3 points: The response draws a valid conclusion supported by the student's data and describes two ways the student could have improved the experimental design and/or the validity of the results;
2 points: The response draws a valid conclusion supported by the student's data and describes one way the student could have improved the experimental design and/or the validity of the results. -or-
The response describes two ways the student could have improved the experimental design and/or the validity of the results but fails to draw or incorrectly draws a conclusion from the student's data;
1 point: The response draws a valid conclusion supported by the student's data but fails to describe, or incorrectly describes, how the student could have improved the experimental design and/or the validity of the results. -or-
The response describes one way the student could have improved the experimental design and/or the validity of the results but fails to draw or incorrectly draws a conclusion from the student's data.;
0 points: The response provides little or no correct information from the polymer investigation.

## C.3 Subset #5

[Question]:
Starting with mRNA leaving the nucleus, list and describe four major steps involved in protein synthesis.

[Key Elements]:
mRNA exits nucleus via nuclear pore.
mRNA travels through the cytoplasm to the ribosome or enters the rough endoplasmic reticulum.
mRNA bases are read in triplets called codons (by rRNA).
tRNA carrying the complementary (U=A, C+G) anticodon recognizes the complementary codon of the mRNA.
The corresponding amino acids on the other end of the tRNA are bonded to adjacent tRNA's amino acids.
A new corresponding amino acid is added to the tRNA.
Amino acids are linked together to make a protein beginning with a START codon in the P site (initiation).
Amino acids continue to be linked until a STOP codon is read on the mRNA in the A site (elongation and termination).

[Rubric]:
3 points: Four key elements;
2 points: Three key elements;
1 point: One or two key elements;
0 points: Other.

## C.4 Subset #6

[Question]:
List and describe three processes used by cells to control the movement of substances across the cell membrane.

[Key elements]:
Selective permeability is used by the cell membrane to allow certain substances to move across.
Passive transport occurs when substances move from an area of higher concentration to an area of lower concentration.
Osmosis is the diffusion of water across the cell membrane.
Facilitated diffusion occurs when the membrane controls the pathway for a particle to enter or leave a cell.
Active transport occurs when a cell uses energy to move a substance across the cell membrane, and/or a substance moves from an area of low to high concentration, or against the concentration gradient.
Pumps are used to move charged particles like sodium and potassium ions through membranes using energy and carrier proteins.
Membrane-assisted transport occurs when the membrane of the vesicle fuses with the cell membrane forcing large molecules out of the cell as in exocytosis.
Membrane-assisted transport occurs when molecules are engulfed by the cell membrane as in endocytosis.
Membrane-assisted transport occurs when vesicles are formed around large molecules as in phagocytosis.
Membrane-assisted transport occurs when vesicles are formed around liquid droplets as in pinocytosis.
Protein channels or channel proteins allow for the movement of specific molecules or substances into or out of the cell.

[Rubric]:
3 points: Three key elements;
2 points: Two key elements;
1 point: One key element;
0 points: Other.