# OpenReview forum: "Distilling ChatGPT for Explainable Automated Student Answer Assessment"
_EMNLP/2023/Conference — EMNLP 2023 Findings_

### Official Review · Reviewer_FgJs · 2023-08-02

**Soundness:** 4

**Excitement:**

3: Ambivalent: It has merits (e.g., it reports state-of-the-art results, the idea is nice), but there are key weaknesses (e.g., it describes incremental work), and it can significantly benefit from another round of revision. However, I won't object to accepting it if my co-reviewers champion it.

**Paper Topic And Main Contributions:**

This paper proposes the AERA framework, which is meant to help both score student answers and generate score rationales. The framework consists of three steps: (1) prompt ChatGPT to generate rationales, (2) refine the generation step to enhance the rationale quality, and (3) fine-tune a smaller LLM using the generated rationales. The authors find that their framework is selected over ChatGPT in rationale correctness and preference, and that the distilled model is mostly better than ChatGPT regarding score prediction.

**Questions For The Authors:**

A)	Lines 186-189: Would this prompting strategy work for models other than ChatGPT? Moreover, what if the underlying model/interface of ChatGPT changes?

B)	Lines 333-337: I am somewhat skeptical of this approach, especially with the cited prior study only being a pre-print. What is the intuition or justification behind this approach? And why do you think it will hold here?

C)	Lines 352-353: Again, I am skeptical of this claim. Do you have any justification for why this should be the case?

D)	Lines 428-429: Why are only four subsets used?

E)	Table 1: This table only reports the classification performance, correct?

F)	Table 1: Was significance testing performed for the comparisons?

G)	Lines 824-839: For reproducibility, it would be helpful to include the learning rate, stopping criteria, warm-up, etc. Moreover, it may be better to do a more thorough hyperparameter search for fairer model comparison.

H)	Appendix Section B2: My concern here is that this doesn’t necessarily tell us about the quality of the rationale or the semantic relation between the rationale and the score. Hypothetically, the rationale could just be the score itself, which would result in high performance reported in A2. Is there any additional justification beyond just the prediction of the score given the rationale?

I)	Appendix Section B6: How frequently was this observed to happen?

**Reasons To Accept:**

The strengths of this paper are as follows:

-	A demonstration of the ability to distill large language model (ChatGPT) for training smaller language models in the domain of education.

-	Positive results for distillation, refinement, and improvement over the original ChatGPT model.

**Reasons To Reject:**

The weaknesses of this paper are as follows:

-	The use of ChatGPT, with it being a closed-source, commercial product, makes it difficult to replicate/reproduce findings and brings into question the generality of the first two steps of the AERA framework. I would suggest repeating the study while using an open-source LLM for replicability and reproducibility, if possible.

-	A lack of significance testing for the qualitative results.

-	The treatment of quality measurement for generated rationales for the second stage (refinement) is a bit confusing.

**Reproducibility:**

3: Could reproduce the results with some difficulty. The settings of parameters are underspecified or subjectively determined; the training/evaluation data are not widely available.

**Reviewer Confidence:**

2: Willing to defend my evaluation, but it is fairly likely that I missed some details, didn't understand some central points, or can't be sure about the novelty of the work.

**Typos Grammar Style And Presentation Improvements:**

-	Figure 2: The text in this figure is very small and therefore difficult to read without zooming in. Would it be possible to make the text or figure larger?

-	Lines 895-898: Is the sentence “Indicating that …” supposed to be a part of the previous sentence?

---

> ### Author Rebuttal · Authors · 2023-08-27
>
> Thank you for your review.
>
> **Re A):** Indeed, the prompting strategy still works for models other than ChatGPT, such as Bard or FlanT5. Our experimentation centred around ChatGPT primarily due to its robust capabilities and cost-efficiency during the experiment design phase. As highlighted by Kaplan, Jared, et al. (2020), larger language models may generally have better performance, and ChatGPT is larger than other open-source LLMs in our experiment phase. The largest open-sourced model we’ve tried was the LLaMA 2 70B Chat model. However, the model struggled to generate coherent rationales in their entirety and tended to repeat the marking rubrics in its response.
>
> Here are the assessment results generated by various models on the following student answer with the Example Instruction prompt “*1. tRNA tranfers it outside the nucleus to be matched with a ribosome2. The ribosome latches on to an amino acid.3. The codons match up4. It creates protein.*”
>
> - **ChatGPT**: 1 point; The student answer only matches one key element, “...tRNA transfers it outside the nucleus to be matched with a ribosome...”.
> - **GPT-4**: 1 point; The student answer matches two key elements, "The codons match up" and "It creates protein".
> - **Google Bard**: The student answer matches only two key elements, ‘tRNA tranfers it outside the nucleus to be matched with a ribosome’ and ‘The codons match up’, so the score is 1 point.
> Here are the four major steps involved in protein synthesis, starting with mRNA leaving the nucleus:
> Initiation: mRNA leaves the nucleus and binds to a ribosome in the cytoplasm. The ribosome then scans the mRNA for a start codon (AUG).
> Elongation: A tRNA molecule with an amino acid attached to it binds to the ribosome at the A site. The anticodon of the tRNA matches the start codon on the mRNA. The amino acid is then added to the growing polypeptide chain. The ribosome then moves one codon down the mRNA.
> Termination: The ribosome continues to read the mRNA until it reaches a stop codon. A release factor binds to the stop codon and the polypeptide chain is released from the ribosome.
> Folding and modification: The newly synthesized polypeptide chain is folded into its correct shape. It may also be modified by the addition of chemical groups.
> The student answer does not mention the initiation step, which is a critical step in protein synthesis. Therefore, the answer is not complete and the score is 1 point.
>
> - **Flan T5 XXL** *(Exceeded max input sequence length)*: 1 point; The student answer matches one key element, “...tRNA tranfers it outside the nucleus to be matched with a ribosome2. The ribosome latches on to an amino acid3. The codons match up4. It creates protein.”.
>
> - **LLaMA 2 70B Chat HF**: Sure, here's the scoring rubric for the student answers:
> [Student Answer]: 1. tRNA tranfers it outside the nucleus to be matched with a ribosome2. The ribosome latches on to an amino acid.3. The codons match up4. It creates protein.
> [Rationale and Score]:
> The student answer matches two key elements: \"tRNA tranfers it outside the nucleus to be matched with a ribosome\" (step 1) and \"It creates protein\" (step 4).* However, the student answer does not mention the other two key elements: \"mRNA bases are read in triplets called codons (by rRNA)\" (step 2) and \"Amino acids are linked together to make a protein beginning with a START codon in the P site (initiation)\" (step 3).
> Score: 2 points
> Here's the corrected scoring rubric:* 4 points: Four key elements;* 3 points: Three key elements;* 2 points: Two key elements;* 1 point: One key element;* 0 points: Other.
>
> **Re B):** The cited prior study has been accepted at the ACL 2023 conference. Due to the absence of verified ground truth rationale data, we posit that the generated rationale supports the assigned score. As far as we are aware, there exists no evaluation metric capable of concurrently assessing both the rationale and the predicted score. Consequently, we proceed with the assumption that the rationale underpins the score, aligning with the prevalent trend observed. Human evaluation and correlation analysis are subsequently provided to validate this assumption as shown in paper 4.3, Appendix B.2, B.3 and B.5, respectively.
>
> **Re C):** We implemented the NLI-based semantic equivalence pipeline from the mentioned prior work. While the majority of the rationales with the same assessed score are categorised as “entail” or “neutral”, it is noteworthy that the remaining instances labelled as “contradict” do not truly exhibit contradictory characteristics. Here are some examples of rationales labelled as “contradict” by the pipeline:
>
> - **Pair 1:**
> 1 point; This response describes one additional piece of information that would be needed to accurately replicate the experiment: “identify the control.” However, the other two pieces of information mentioned, the type of material and the unit of measurement for mass, were already provided in the procedure.
> 1 point; This response describes one additional piece of information that would be needed to accurately replicate the experiment: “identify the control.” However, the other two pieces of information mentioned are not necessary to replicate the experiment as the type of material and the unit of measurement are already provided in the procedure.
>
> - **Pair 2:**
> 2 points; This response describes two additional pieces of information that would be needed to accurately replicate the experiment: “how much vinegar you would put the samples in” and “how long do you rinse the samples for.” However, the second point is unclear and could benefit from clarification.
> 2 points; This response describes two additional pieces of information that would be needed to accurately replicate the experiment: “how much vinegar you would put the samples in” and “how long do you rinse the samples for.” However, the second point is already answered in step 4 of the procedure, so it is not a necessary additional piece of information.
>
> Therefore, we decided to group the noise rationales by the assessed score.
>
> **Re D):** We focus on those Science subsets that offer key answer elements, and are accompanied by rubrics, thus rendering the assessment process a clear reasoning task. Conversely, English and Arts questions from the dataset adopt an Open-ended format and include multi-modal data, which require additional knowledge to make objective judgements.
>
> **Re E):** Yes, the table only reports classification performance since we don’t have the gold rationale data. Optimal checkpoints for rationale generation models are selected based on their BLEU score in comparison to the ChatGPT-generated rationales. Rationales are evaluated by human annotators.
>
> **Re F):** Yes, significance testing was conducted for the comparisons. Specifically, we performed one-tailed t-tests to assess the performance of AERA in comparison with various ablation distilled baselines. The findings are summarized as follows:
>
> The P-value comparing AERA and the 'w/o Fixing Wrong Labels' baseline is 0.002, indicating a statistically significant improvement with AERA.
>
> Similarly, the P-value when comparing AERA to the 'w/o Rationale Refinement' baseline is 0.011, again signifying that AERA exhibits a significant improvement over models that do not incorporate this specific refinement strategy.
>
> For the comparison between AERA and ChatGPT, the P-value is 0.1455. The relatively higher P-value can be attributed to ChatGPT's underperformance in test #2, as elaborated upon in our "Overall Comparison" section.
>
> Lastly, the P-value for the comparison between AERA and the 'Correct Score Only' baseline stands at 0.1080.
>
> In summary, AERA generally displays statistically significant improvements over models excluding specific refinement strategies. However, for certain comparisons like AERA vs. ChatGPT and AERA vs. 'Correct Score Only', the improvements, though noteworthy, did not reach conventional levels of statistical significance.
>
> **Re G):** We used learning rates of 1e-5 and 1e-4 for the classification and generation tasks respectively. We set a weight decay of 0.01. We didn’t apply any warmup or stopping criteria. All the runs were concluded within the specified epoch count, and the best checkpoints were chosen by the BLEU score. We will conduct a more thorough  hyperparameter search as you suggested. Regarding your reproducibility concerns, we will release both the source code and all the query results from the ChatGPT for the sake of ensuring reproducibility.
>
> **Re H):** The generated output of our model follows a format of "x point(s); (explanation based on key elements and rubric)", which we have defined in the method part as "$\hat{y}_i$; $\hat{r}_i$". In the experiment outlined in Appendix B.2, we removed the assessed score ($\hat{y}_i$), retaining solely the rationales ($\hat{r}_i$) following the semicolon. This step ensures that the classifier is entirely precluded from direct access to the score.
>
> **Re I):** These cases are studied based on manually picked 45 hallucination cases from 6542 complex instruction queried results.
>
> **Re** *Typos Grammar Style And Presentation Improvements*
>
> Thank you for your advice. We will adjust the figure size and correct the sentence in the revised version.
>
> **References:**
> Kaplan, Jared, et al. "Scaling laws for neural language models." arXiv preprint arXiv:2001.08361 (2020).

---

### Official Review · Reviewer_PxdN · 2023-08-04

**Soundness:** 4

**Excitement:**

3: Ambivalent: It has merits (e.g., it reports state-of-the-art results, the idea is nice), but there are key weaknesses (e.g., it describes incremental work), and it can significantly benefit from another round of revision. However, I won't object to accepting it if my co-reviewers champion it.

**Paper Topic And Main Contributions:**

This paper introduces a novel framework - Automated Explainable Student Response Assessment (AERA), for automated student answer assessment, which utilizes ChatGPT, a cutting-edge large language model, for both scoring student answers and generating rationales. The approach entails prompting ChatGPT with different templates to obtain rationales, which are then refined to align with standard marking guidelines in cases of inconsistency. The refined outputs from ChatGPT are used to fine-tune a smaller language model - Long T5, which simultaneously evaluates student responses and provides corresponding rationales. The paper demonstrates through extensive experiments on a benchmark dataset that their method improves the overall Quadratic Weighted Kappa (QWK) score by 11% compared to ChatGPT alone. Further, a thorough automatic and human evaluation show that the rationales generated by the proposed method are of comparable quality to those produced by ChatGPT. AERA offers a practical solution for achieving explainable and automated assessment in the field of education.

**Questions For The Authors:**

Question A: In the example instruction setting and Prompt for Rational Refinement setting, how many demonstration examples were used within the few-shot setting for ChatGPT?

Question B: Following the above question. Did you experiment with varying the number of demonstration examples and subsequently compare the performance results?

Question C: The dataset (ASAP-SAS) you used doesn't include the rationale for each score. What are the ground truths for evaluating the generated rationales by Long T5 and ChatGPT?

Question D: How much are the running costs of the whole evaluation with ChatGPT API?

**Reasons To Accept:**

1. The proposed framework has the potential to greatly influence the education field by making student answer assessments more efficient, consistent, and interpretable. This could lead to improved teaching and scoring mechanisms and enhance learning outcomes, potentially revolutionizing the way educators approach grading and feedback.

2. The paper is appreciated for its clear and fluent writing style. The authors effectively communicate the overall idea, experiment setting, and results, which helps understanding thoroughly of their proposed method and findings.

**Reasons To Reject:**

1. The rationale for choosing the ASAP-SAS dataset is not clear, especially given that this dataset lacks rationales for each score, which might limit the evaluation of the generated rationales.

2. There is no clear mention of comparison with varying the number of demonstration examples in the experiments, which could provide valuable insights into the model's performance.

**Reproducibility:**

4: Could mostly reproduce the results, but there may be some variation because of sample variance or minor variations in their interpretation of the protocol or method.

**Reviewer Confidence:**

3: Pretty sure, but there's a chance I missed something. Although I have a good feel for this area in general, I did not carefully check the paper's details, e.g., the math, experimental design, or novelty.

---

> ### Author Rebuttal · Authors · 2023-08-27
>
> Thank you for your insightful review, and acknowledge our contribution to the education domain.
>
> **Re Reject 1:**
>
> “*The rationale for choosing the ASAP-SAS dataset is not clear…*”
>
> To the best of our knowledge, a science student answer dataset with publicly available rationales explaining human-assigned scores does not currently exist. This serves as the driving force behind our proposed AERA framework. This framework aims to refine the somewhat noisy rationales generated by ChatGPT, using them to train a smaller model capable of both assessing answers and generating rationales. Our selection of the ASAP-SAS dataset is underpinned by a few key reasons:
>
> - We place particular emphasis on evaluating answers in scientific subjects, and the ASAP-SAS dataset conveniently encompasses four subsets that revolve around Science and Biology questions.
> - The problem structure within ASAP-SAS, wherein student responses are paired with key answer components and then assessed using a grading rubric to determine the ultimate score, closely mirrors the human scoring process employed in real-world scenarios. This likeness suggests that methodologies trained on this dataset can be readily adapted to real examination systems.
>
> **Re Reject 2:**
>
> “*Comparison with varying the number of demonstration examples…*”
>
> We didn’t make a comparison with varying the number of demonstration examples. We agree to add an ablation study on how different amounts of demonstrations may affect the instruction performance. The analysis of the influence of few-shot performance on the number of demonstrations and selection of demonstrations has been widely researched by [Brown, Tom, et al 2020; Zhang, Zhuosheng, et al 2022], and a common agreement is that more demonstration examples that cover a wide range of scenarios provide better few-shot performance. As mentioned in Re Reject 1, the examination standardisation meeting will provide a few benchmark student answers from all score ranges. Therefore, we expect the user of our framework to adopt as many of those examples as possible. We have conducted the comparison of few-shot performance on the number of demonstration examples in the reply to **Question B**.
>
> **QuestionA:** “*How many demonstration examples were used within the few-shot setting for ChatGPT?*”
>
> This varies by how many demonstrations are available from the ASAP-SAS dataset. There are 8 examples for #1 and #2, 6 examples for #5, and 5 examples for #6. Those samples cover all score ranges.
>
> **Question B:** “*experiment with varying the number of demonstration examples*”
>
> Following your advice, we have performed an ablation study with the latest ChatGPT version on the influence of the number of demonstration samples on #6, as shown in the table below:
> | Number of demonstrations | ChatGPT Version | Scores for demonstrations | Accuracy | F1 | QWK |
> |---|---|---|---|---|---|
> | 5 | 23 Mar 2023, reported in the paper | 0,1,2,3,3 | 68.45 | 50.39 | 64.66 |
> | 5 | 3 Aug 2023 | 0,1,2,3,3 | 79.97 | 54.66 | 71.48 |
> | 4 | 3 Aug 2023 | 0,1,2,3 | 77.96 | 40.75 | 68.63 |
> | 3 | 3 Aug 2023 | 0,1,2 | 74.62 | 48.75 | 66.24 |
> | 2 | 3 Aug 2023 | 0,1 | 72.62 | 42.82 | 54.90 |
> | 1 | 3 Aug 2023 | 0 | 67.45 | 39.30 | 57.26 |
>
> Similar to observations reported in prior work [Brown, Tom, et al., 2020], the performance drops with fewer demonstration examples. We also observed the same trend on other subsets.
>
> **Question C:** “*What are the ground truths for evaluating the generated rationales by Long T5 and ChatGPT*”
>
> There is no ground truth rationale. So, we hired human annotators to evaluate the faithfulness of the rationale on both Long T5 and ChatGPT-generated outputs.
>
> **QuestionD:** “*the running costs of the evaluation with ChatGPT API*”
>
> The majority of the experiments were carried out before the ChatGPT token prices were reduced. One run for a single dataset cost around $3.33.
>
> **References:**
> Brown, Tom, et al. "Language models are few-shot learners." Advances in neural information processing systems 33 (2020): 1877-1901.
> Zhang, Zhuosheng, et al. "Automatic chain of thought prompting in large language models." arXiv preprint arXiv:2210.03493 (2022).

---

### Official Review · Reviewer_JJVy · 2023-08-06

**Soundness:** 2

**Excitement:**

3: Ambivalent: It has merits (e.g., it reports state-of-the-art results, the idea is nice), but there are key weaknesses (e.g., it describes incremental work), and it can significantly benefit from another round of revision. However, I won't object to accepting it if my co-reviewers champion it.

**Paper Topic And Main Contributions:**

The paper develops a distillation method for Short Answer Grading by fine-tuning a smaller model on ChatGPT explanations. The method consists of three steps: (1) generating rationales from ChatGPT, (2) refining those explanations, and (3) fine-tuning a Longformer T5 model on the refined explanations (and labels). The paper conducts experiments on a short answer grading dataset, specifically, ASAP-SAS to show that the distilled model outperforms ChatGPT and generates explanations that are comparable to ChatGPT.

**Reasons To Accept:**

* The paper focuses on an important and a long-standing application of NLP, that of explainable short answer grading.

* The proposed distillation method is simple and the use of a bigger model for generating explanations to overcome data scarcity issue in short answer grading is interesting. I also liked the refinement prompt, that conditions on the label to improve the rationales. The ablations also show that it helps improve the explanations from ChatGPT.


**Reasons To Reject:**

* **Weak Experiments**: The paper compares their distilled model AERA (that generates both scores and rationales) to (1) ChatGPT (which also generates both), and (2) some classification models that only output scores. In my opinion, the conclusions drawn from these baselines are weak. The classification models, while they do not generate rationales, obtain an F1 score which is about 9-10 points better than the AERA model (64 vs 55). I understand the accuracy-explainability tradeoff but this is quite a significant drop. On the other hand, comparison to ChatGPT is also somewhat unfair when the experiments are performed on the same dataset for which the model has been tuned. ChatGPT is a zero-shot model while AERA is fine-tuned on a lot of filtered demonstrations from ChatGPT. So the paper needs to experiment on other datasets, for which ChatGPT demonstrations are not available/used to show that their distilled model generalizes.

* **Lack of Novelty**: While the application of short answer grading may be new, the idea of fine-tuning a smaller model with the explanations of a larger model is not novel. The work should at least discuss these papers and talk about how their method differs from these existing distillation papers (and there are quite a lot of them). Listing a few below.

[1] Teaching small language models to reason. Magister et al. 2023
[2]  Specializing smaller language models towards multi-step reasoning. Fu et al. 2023
[3] Large language models are reasoning teachers. Ho et al. 2023

**Reproducibility:**

4: Could mostly reproduce the results, but there may be some variation because of sample variance or minor variations in their interpretation of the protocol or method.

**Reviewer Confidence:**

4: Quite sure. I tried to check the important points carefully. It's unlikely, though conceivable, that I missed something that should affect my ratings.

---

> ### Author Rebuttal · Authors · 2023-08-27
>
> Thank you for your review.
>
> **Re Weak Experiments:**
>
> “*Big performance gap between classification model and AERA.*”
>
> As outlined in the 4.2 Overall Comparison and illustrated in Table 1, our observations reveal that the Longformer classifier exhibits superior performance exclusively with student answers (Longformer), in contrast to cases where student answers are coupled with essential assessment details like questions, key elements, and rubrics (Longformer-all). This outcome suggests that a text classifier-based answer assessment system falls short of simulating human grading behaviours and potentially suffers from spurious correlation. The empirical cases discussed in Section 4.4 also point to the potential corruption of some data or inconsistencies among annotators. Thus, the text classifier-based assessment system inflates its apparent effectiveness.
>
> Therefore, we are motivated to develop a more interpretable assessment system due to this lack of trustworthiness. It is important to note, however, that our framework, as declared in the introduction, does not aim to outperform the existing classifier approach in terms of raw performance. Our focus centres on reframing the current approach to answer assessment by enhancing interpretability through the generation of rationales.
>
> Lastly, it's worth noting that the latest ChatGPT version, as of 05 August 2023, has demonstrated significant improvement across various metrics. This progression underscores the potential for AERA to bridge the current performance differential.
>
> “*Experiment on other datasets where ChatGPT demonstrations are not available…*”
>
> Our objective is to develop interpretable student assessment systems to make the application more trustworthy. Nonetheless, due to the absence of a comprehensively annotated dataset offering feedback on student responses related to science, we find it necessary to produce rationales through distillation. This process mandates the utilisation of LLM's demonstrations, as comparing few-shot LLM’s results and fine-tuned smaller language model results have been frequently employed in your suggested literature as well.
>
> Unlike the filtering strategy employed in previous endeavours, the refinement approach within AERA *did not exclude* any instances of ChatGPT demonstrations from the training set. The evaluation result is solely based on the test set, which remained untouched during the training phase. As a result, the test outcomes present an equitable comparison, with both ChatGPT and AERA solely engaged based on inference.
>
> **Re Lack of Novelty:**
>
> While previous attempts have involved fine-tuning a smaller model using rationales generated from a larger one, our achievement lies in being the pioneers of a workable solution for explainable student answer scoring.
> As highlighted in [EMNLP 2020, 2020], we suggest that our work has made both scientific and engineering contributions:
>
> *Scientific contributions*:
> 1. Our straightforward yet effective refinement strategy has notably enhanced the performance of the distilled model by leveraging LLM’s in-context learning capability.
> 2. Our framework stands out as the first to improve interpretability using LLMs to generate rationales for assessing student answers.
>
> *Engineering contributions*:
> 1. Our refinement strategy offers a novel means of identifying corrupted data through an assessment of LLM's predictive uncertainty, which hasn’t been explored in prior works to the best of our knowledge. This strategy significantly mitigates the impact of noisy data annotations, which can hardly be discovered via traditional classification approaches.
> 2. Our framework simulates human marking behaviours, and framework workflow can be easily adapted to real-world examination scenarios.
>
> For the suggested related literature, we have cited [3] and discussed it in our paper, and we will discuss the others in the revised version.
>
> **Reference**:
> EMNLP 2020 (2020) How to write good reviews, EMNLP 2020. Available at: https://2020.emnlp.org/blog/2020-05-17-write-good-reviews (Accessed: 28 August 2023).

---

### Meta-Review · Area_Chair_zPqV · 2023-09-20

**Recommendation:** 3

**Metareview:**

This paper introduces a novel framework called Automated Explainable Student Response Assessment (AERA), which is capable of distilling ChatGPT as a reasoning teacher to fine-tune a smaller language model, generating score rationales, and improving the interpretability on student answer assessment tasks. The proposed method consists of three steps: (1) generating rationales using various prompt templates with ChatGPT, (2) refining output explanations, and (3) fine-tuning a Long T5 model on these refined explanations and labels. In addition, as part of this process, the quality of rationales is enhanced with a rationale refinement module. The authors run extensive experiments, which show that a smaller language model can surpass ChatGPT in terms of assessment performance while generating more accurate rationales to explain the assessment decisions. Specifically, the proposed method improves the overall Quadratic Weighted Kappa (QWK) score by 11% compared to ChatGPT alone, and a human evaluation confirms that the rationales generated by the new method are of comparable quality to those produced by ChatGPT.

All reviewers highlight **a number of strengths and merits** of this paper:
1. *Motivation*: Reviewer JJVy mentions that the paper focuses on an important and a long-standing application of NLP. In addition, Reviewer PxdN believes that the proposed framework has the potential to influence the education field by making student answer assessments more efficient, consistent, and interpretable.
2. *Methodology*: Reviewer JJVy finds the proposed distillation method simple but effective and the rationale refinement module interesting. Reviewer FgJs agrees with this view.
3. *Evaluation*: Reviewers JJVy and FgJs also find ablation studies useful and informative.
4. *Clarity of writing and presentation* is listed among the reasons to accept by Reviewer PxdN.

At the same time, reviewers identify **further weaknesses of the current submission** and areas for improvement:
1. *Concerns about the choice of the dataset* are expressed by reviewers JJVy and PxdN (please see their reviews for more details). While the ideal dataset for this task may not yet exist, these concerns should be addressed in the revision.
2. *Concerns about the experimental setup*: Reviewer JJVy has posed a number of questions about the setup and comparison of the proposed model to the baselines. The authors have tried to address some of these concerns – these answers should be incorporated into the revision. For instance, there does seem to be a trade-off between the performance of the models and their ability to provide rationales, and while it is the latter that was the main focus of this work, this trade-off should be addressed in the paper. This may require comparison to more baselines. In addition, presence of spurious correlations, which contribute to certain baselines' performance, is currently hypothesized in the paper – this needs to be supported by stronger evidence and possibly additional experiments. Please also see further suggestions on the experimental setup from Reviewer PxdN.
3. *Concerns about the use of ChatGPT* are expressed, to various extent, by all reviewers. Reviewer FgJs, in particular, suggests running experiments with open-source LLMs for replicability and reproducibility. The question of generalizability of the findings is also raised by more than one reviewer.
4. *Other suggestions on the presentation of evaluation results* (such as inclusion of significance testing) are expressed by Reviewer FgJs.
5. *Missing references* are also suggested by Reviewer JJVy.

---

### Decision · Program_Chairs · 2023-10-07

**Decision:**

Accept-Findings

**Comment:**

This paper introduces a novel framework called Automated Explainable Student Response Assessment (AERA), which is capable of distilling ChatGPT as a reasoning teacher to fine-tune a smaller language model, generating score rationales, and improving the interpretability on student answer assessment tasks. The proposed method consists of three steps: (1) generating rationales using various prompt templates with ChatGPT, (2) refining output explanations, and (3) fine-tuning a Long T5 model on these refined explanations and labels. In addition, as part of this process, the quality of rationales is enhanced with a rationale refinement module. The authors run extensive experiments, which show that a smaller language model can surpass ChatGPT in terms of assessment performance while generating more accurate rationales to explain the assessment decisions. Specifically, the proposed method improves the overall Quadratic Weighted Kappa (QWK) score by 11% compared to ChatGPT alone, and a human evaluation confirms that the rationales generated by the new method are of comparable quality to those produced by ChatGPT.

All reviewers highlight **a number of strengths and merits** of this paper:
1. *Motivation*: Reviewer JJVy mentions that the paper focuses on an important and a long-standing application of NLP. In addition, Reviewer PxdN believes that the proposed framework has the potential to influence the education field by making student answer assessments more efficient, consistent, and interpretable.
2. *Methodology*: Reviewer JJVy finds the proposed distillation method simple but effective and the rationale refinement module interesting. Reviewer FgJs agrees with this view.
3. *Evaluation*: Reviewers JJVy and FgJs also find ablation studies useful and informative.
4. *Clarity of writing and presentation* is listed among the reasons to accept by Reviewer PxdN.

At the same time, reviewers identify **further weaknesses of the current submission** and areas for improvement:
1. *Concerns about the choice of the dataset* are expressed by reviewers JJVy and PxdN (please see their reviews for more details). While the ideal dataset for this task may not yet exist, these concerns should be addressed in the revision.
2. *Concerns about the experimental setup*: Reviewer JJVy has posed a number of questions about the setup and comparison of the proposed model to the baselines. The authors have tried to address some of these concerns – these answers should be incorporated into the revision. For instance, there does seem to be a trade-off between the performance of the models and their ability to provide rationales, and while it is the latter that was the main focus of this work, this trade-off should be addressed in the paper. This may require comparison to more baselines. In addition, presence of spurious correlations, which contribute to certain baselines' performance, is currently hypothesized in the paper – this needs to be supported by stronger evidence and possibly additional experiments. Please also see further suggestions on the experimental setup from Reviewer PxdN.
3. *Concerns about the use of ChatGPT* are expressed, to various extent, by all reviewers. Reviewer FgJs, in particular, suggests running experiments with open-source LLMs for replicability and reproducibility. The question of generalizability of the findings is also raised by more than one reviewer.
4. *Other suggestions on the presentation of evaluation results* (such as inclusion of significance testing) are expressed by Reviewer FgJs.
5. *Missing references* are also suggested by Reviewer JJVy.